# TransMarker: Unveiling dynamic network biomarkers in cancer progression through cross-state graph alignment and optimal transport

Fatemeh Keikha[1], Chuanyuan Wang[1], Zhixia Yang[2], Zhi-Ping Liu[1,2,3]*

**1** Department of Biomedical Engineering, School of Control Science and Engineering, Shandong University, Jinan, Shandong, China, **2** College of Mathematics and Systems Science, Xinjiang University, Urumqi, Xinjiang, China, **3** National Center for Applied Mathematics, Shandong University, Jinan, Shandong, China

* zpliu@sdu.edu.cn

## Abstract

The identification of state specific biomarkers that reflect dynamic changes in gene regulatory networks is critical for understanding cancer progression and enhancing diagnostic precision. While multilayer network models have been proposed for analyzing disease evolution, most existing methods rely solely on topological features, neglecting structural rewiring and expression variability across disease states. In this study, we introduce TransMarker, a framework designed to detect genes with regulatory role transitions, those with meaningful shifts in regulatory roles during disease progression, as dynamic biomarkers via cross-state alignment of multi-state single-cell data. TransMarker encodes each disease state as a distinct layer in a multilayer graph, integrating prior interaction data with state-specific expression to construct attributed gene networks. Contextualized embeddings for each stage are generated for each state using Graph Attention Networks (GATs), and structural shifts are quantified via Gromov-Wasserstein optimal transport. Genes with significant changes are ranked using a Dynamic Network Index (DNI), which captures their regulatory variability. These prioritized biomarkers are then applied in a deep neural network for disease state classification. We validate our approach on synthetic simulated and real world dataset of gastric adenocarcinoma (GAC), to evaluate performance across diverse scenarios and assess generalizability. TransMarker outperforms existing multilayer network ranking techniques in classification accuracy, robustness, and biomarker relevance. Ablation studies confirm the contribution of each step to overall performance. Our findings suggest that combining regulatory rewiring, temporal expression dynamics, and cross-state alignment provides a powerful strategy for identifying biologically meaningful biomarkers and modeling disease progression at single cell resolution.

**Data availability statement:** The source code and data are available at http://www.github.com/zpliulab/TransMarker.

**Funding:** National Natural Science Foundation of China (Grant Nos. 92374107 and 62373216; https://www.nsfc.gov.cn/) to Z.L, National Key Research and Development Program of China (Grant No. 2020YFA0712402; https://service.most.gov.cn/) to Z.L, Shandong Provincial Key Research and Development Program (Major Scientific and Technological Innovation Project) (Grant No. 2023CXGC010509; http://kjt.shandong.gov.cn/) to Z.L. The sponsors or funders had no role in the study design, data collection and analysis, decision to publish, or preparation of the manuscript.

## Author summary

Understanding the progression of cancer from one state to another necessiates identifying pivotal genes whose regulatory roles evolve over time. Traditional approaches often concentrate solely on static gene networks, overlooking the dynamic evolution of these networks across disease states. To bridge this gap, we developed TransMarker, a computational framework that detects genes with shifting regulatory roles by analyzing gene expression and interactions across various disease states using single-cell data. TransMarker models each disease state as a distinct layer in a multilayer network and employs advanced machine learning techniques, including graph attention networks and optimal transport, to quantify changes in gene behavior. We evaluated our method on both simulated data and real single-cell data from gastric cancer, demonstrating that TransMarker can accurately classify disease states and pinpoint meaningful biomarkers. Our findings indicate that accouting for temporal evolution of gene regulation enhances our capacity to monitor disease progression.

## Introduction

Cancer is inherently complex, driven by both genetic and environmental factors that interact through diverse biological mechanisms [1]. Its initiation and progression cannot be solely attributed to alterations in DNA sequences or gene expression levels; rather, growing evidence suggests that disruptions in gene-gene interactions also play a crucial role [2]. In fact, recent studies highlight that changes at the interaction level changes may be as significant as individual genetic alterations in shaping cancer dynamics [3]. Notably, many cancer associated mutations occur in only a small subset of patients or are specific to certain cancer types or subtypes, leading to their underrepresentation in traditional genetic analyses [4]. To better understand these complexities, some researchers employ evolutionary frameworks to model how genetic mutations and network perturbations can drive disease progression over time.

However, cancer is not a static condition; it progresses through distinct pathological states, each characterized by dynamic shifts in the molecular and cellular landscape [5]. These transitions are often driven not only by accumulated mutations but also by changes in regulatory interactions and signaling dynamics that govern cellular behavior [6–8]. Crucially, the alterations mediating transitions between disease states, such as from normal to precancerous or from localized to metastatic, may be subtle, rare, or transient, making them difficult to capture through static or bulk analyses [9]. Identifying state-specific molecular switches that orchestrate these shifts is essential for understanding cancer progression and discovering precise intervention points [10,11]. This necessitates computational approaches capable of detecting dynamic biomarkers and network rewiring events that reflect these critical transitions [12].

Recent studies have demonstrated the value of co-expression and regulatory network analyses in elucidating the molecular underpinnings of cancer by capturing

changes in gene-gene correlations across contexts such as normal versus tumor tissues, distinct cancer types, and molecular subtypes [13–18]. While stable gene interactions often represent core housekeeping processes, context specific or newly formed connections are frequently linked to oncogenic mechanisms and disease specific perturbations [2]. Traditional network based gene prioritization methods, such as those rooted in the guilt by association principle, have shown success in identifying disease associated genes; however, they commonly suffer from limitations including bias toward highly connected (hub) genes, insufficient tissue or state specificity, and inadequate integration of multi modal or temporal data [19,20].

To overcome these issues, recent approaches have begun to leverage dynamic network modeling and deep learning techniques. For instance, RL-GenRisk, a deep reinforcement learning framework, formulates gene prioritization in renal carcinoma as a Markov Decision Process over graph structured data, integrating expression and topology with adaptive rewards to overcome the scarcity of known risk genes, leading to successful identification and validation of ccRCC associated genes [21]. Similarly, DyNDG introduces a time series multilayer network for leukemia, where stage specific DEGs are embedded in a dynamic framework that incorporates both evolving and static biological interactions; the method demonstrates superior performance in identifying progression associated genes by explicitly modeling network rewiring [20]. Complementing these algorithmic advancements, recent single cell atlases of aging tissues, such as those in mammary glands, have revealed coordinated transcriptomic and epigenomic changes across epithelial, stromal, and immune compartments that mirror early tumorigenic transitions emphasizing that molecular rewiring precedes overt transformation and reinforcing the need for dynamic models in cancer biology [22].

Recent studies have extended Dynamic Network Biomarker research by devising advanced frameworks for detecting early-warning and critical-transition signals in complex systems. Methods such as spatiotemporal information transformation learning (RSIT) [23] and spatial–temporal PCA (stPCA) [24] identify system instability and tipping points from high-dimensional data. In disease modeling, the sample-perturbed Gaussian graphical model (sPGGM) [25] and sample-specific causality network entropy (SCNE) [26] uncover pre-disease or pre-deterioration states via individualized network analysis. These approaches emphasize the growing significance of DNB-inspired methods for detecting critical transitions and inspire the design of our framework.

Together, these advances underscore the importance of capturing dynamic changes in network topology over time or disease progression to better understand tumor evolution and identify state relevant biomarkers that static methods may overlook. Such findings motivate the development of models that can explicitly characterize temporal or state specific network reconfiguration, which is central to our proposed approach.

Recently, several dynamical optimal transport (OT)–based frameworks have emerged for modeling temporal cell-state transitions and inferring gene regulatory dynamics from single-cell transcriptomic data. Notable examples include TrajectoryNet [27], which models continuous developmental trajectories using neural OT; TIGON [28], which integrates OT with graph neural networks to uncover temporally evolving regulatory interactions; DeepRUOT [29], which extends regularized unbalanced OT for robust cross-time gene expression alignment; and stVCR [30], which employs stochastic regularization to improve velocity-consistent reconstruction of single-cell trajectories. These approaches collectively highlight the power of OT for capturing dynamic molecular processes and inferring regulatory relationships from continuous transitions. Inspired by these developments, our work extends the OT principle to cross-state network alignment in cancer, focusing on identifying dynamic network biomarkers that reflect regulatory rewiring across discrete pathological states rather than continuous cell transitions.

To address the challenges of capturing dynamic molecular mechanisms in cancer, we introduce TransMarker, a novel framework for identifying Dynamic Network Biomarkers (DNBs) by aligning gene regulatory networks across disease states using single cell expression data. First, our approach encodes each disease state as a separate layer in a multilayer graph, where intralayer edges capture state specific interactions and interlayer connections reflect shared genes across states. Second, we construct enriched regulatory graphs for each state by integrating gene expression data with prior interaction networks, and extract both local and global topological features. These attributed graphs are then passed

through graph attention networks (GATs) to learn contextual embeddings that reflect both within state structure and cross-state dynamics. Instead of aligning networks directly, we leverage Gromov-Wasserstein optimal transport to measure the structural shift of each gene across states in the learned embedding space. Genes with high alignment shifts are treated as candidates, from which all union connected subnetworks are built to compute a DNI that captures structural variability. Genes in a connected subnetwork with the top DNI score are identified as DNBs. Finally, a Deep Neural Network (DNN) uses these DNBs, along with the initial feature matrices, to perform multiclass classification of disease states.

Our framework is specifically designed to detect switcher genes, those with inconsistent yet biologically meaningful behavior across stages, enabling the discovery of biomarkers that play critical roles in disease progression. Unlike traditional multilayer centrality based methods, our approach combines temporal dynamics, structural reconfiguration, and expression variability, resulting in biomarkers with stronger biological relevance. Experimental results on several independent single-cell datasets demonstrate that TransMarker consistently outperforms state of the art multilayer ranking methods in classification accuracy, robustness, and interpretability. Our findings suggest that modeling cross stage network rewiring is crucial for uncovering molecular drivers of cancer progression and advancing precision diagnostics.

The main contributions of our study can be summarized as follows:

- We construct state specific gene regulatory networks by integrating single cell gene expression data with prior biological interactions, enabling context aware modeling of disease progression.
- Our framework formulates disease states as layers in a multilayer graph and leverages both intralayer (local) and interlayer (cross-state) information to capture dynamic molecular shifts.
- We introduce a novel strategy based on Gromov-Wasserstein optimal transport to quantify structural alignment shifts of genes across stagts, facilitating the identification of dynamic switcher genes.
- We propose a DNI that quantifies network variability and enables robust prioritization of dynamic network biomarkers involved in disease transitions.
- Experimental results on three simulated datasets and three real independent datasets validate the effectiveness of our method in identifying biologically relevant biomarkers and demonstrate superior performance in multi-state classification tasks compared to existing multilayer network ranking approaches.

## Results

### TransMarker framework

We propose a novel framework, TransMarker, for identifying Dynamic Network Biomarkers (DNBs) from single cell time course gene expression data using gene regulatory networks. The overall framework of TransMarker is presented in Fig 1.

The approach starts by integrating expression profiles with a prior gene interaction matrix to generate state specific attributed graphs (Fig 1A), where nodes represent genes and edges denote regulatory interactions (Fig 1B). To extract informative features, the framework incorporates both local (shortest path distances) and global (PageRank scores) structural perspectives into the interaction matrices (Fig 1c). These enriched graphs are then passed through GATs to learn contextualized embeddings for each disease state (Fig 1D).

Next, Gromov-Wasserstein (GW) optimal transport is used to compute alignment scores between states, capturing changes in gene network positions over time. Genes with high alignment shifts are selected as candidates for dynamic analysis (Fig 1E). A union network is built from these genes, and a DNI is computed to quantify structural variability. Genes with the highest DNI values are identified as DNBs (Fig 1F).

In the final step, TransMarker uses the identified DNBs along with the Initial Feature Matrices of samples and genes as input to a Deep Neural Network (DNN) for multiclass classification, evaluating the effectiveness of the biomarkers in distinguishing disease states (Fig 1G).

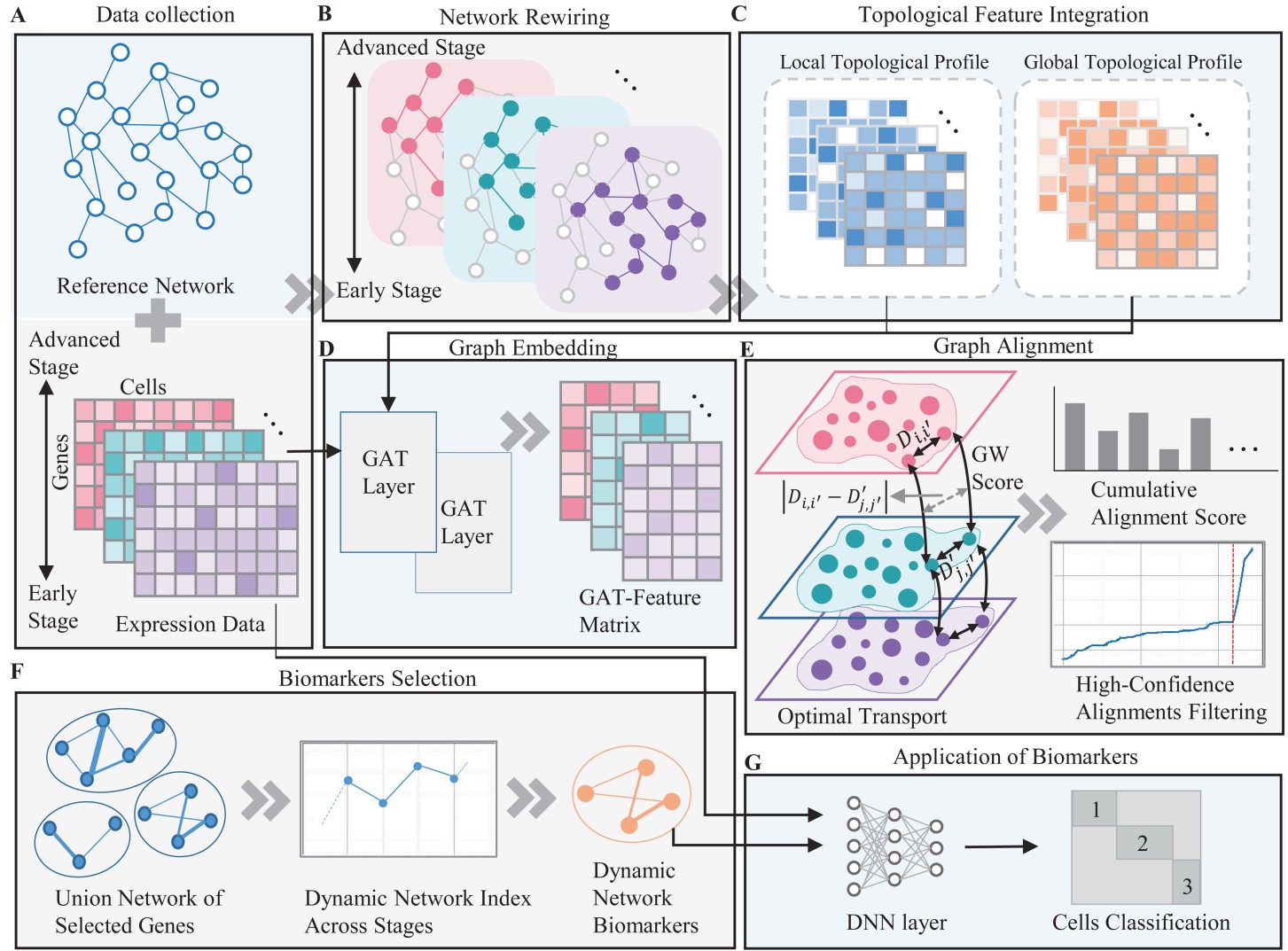

**Fig 1. Overview of the TransMarker framework for identifying DNBs from single cell time course data.** (A) State specific gene expression profiles are integrated with a prior regulatory network to construct attributed gene interaction graphs. (B) A set of rewired graphs is generated to represent disease progression across multiple states. (C) Topological Feature Integration incorporates both local and global structural properties into each graph. (D) GATs are applied to learn contextualized embeddings for each disease state. (E) Gromov-Wasserstein optimal transport computes alignment scores between state specific networks to identify genes with high structural shifts. (F) A union subnetwork of selected genes is used to compute a DNI, identifying DNBs based on network variability. (G) The resulting DNBs and initial feature matrices are input to a DNN for multiclass classification across disease states.

## Performance on simulation datasets

We utilized a single cell simulation framework from the SERGIO toolkit, as introduced by Dibaeinia et al. [31]. This tool models regulatory interactions among multiple transcription factor (TF)–gene pairs based on predefined gene regulatory network (GRN) structures. SERGIO simulates cellular differentiation trajectories using stochastic differential equations and generates gene expression profiles by capturing the system at its steady state. The cell splicing step is then simulated, and several transformations are applied to approximate the actual state of the cell. Lastly, the simulator introduces technical noise, outliers, and dropout effects, and converts the simulated data into UMI count format.

In this study, we simulate gene expression profiles for 30, 50, and 100 genes within a single cell type across four disease states, resulting in three simulation datasets: D1, D2, and D3. The corresponding simulated GRNs are depicted in Fig 2A, 2B, and 2C, representing networks with 30, 50, and 100 genes, respectively. Based on these regulatory interactions, we generate 100 cells per state. For instance, Fig 2D displays the expression profiles of 30 genes over four states, with 100 simulated cells in each state, where each state captures a distinct time course progression.

To evaluate the effectiveness of the proposed framework, TransMarker was initially tested on the three simulation datasets (D1, D2, and D3), across four disease states. These datasets provide a controlled setting to assess how well the model captures dynamic gene expression patterns and identifies state discriminative features.

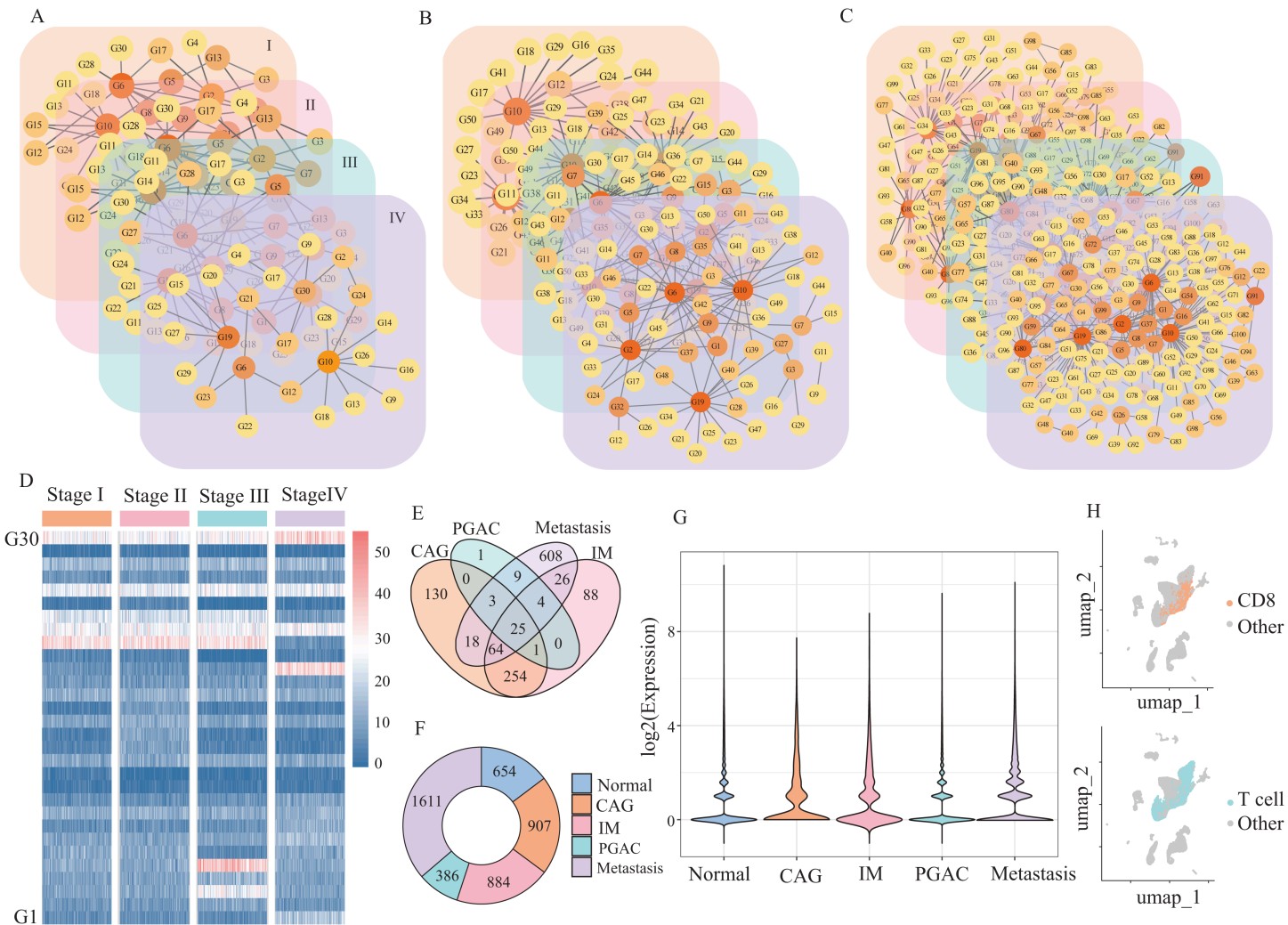

**Fig 2. Simulation networks and gene expression analysis across disease states.** (A-C) Simulated GRNs for datasets D1, D2, and D3, containing 30, 50, and 100 genes, respectively, generated using the SERGIO simulator. (D) Simulated gene expression matrix for dataset D1, showing expression levels of 30 genes across four disease states, with 100 cells per state. (E) Venn diagram showing the overlap of retained edges between the four disease states of GAC after applying the PC-CMI algorithm. (F) Number of DEGs identified across the five conditions (normal and four disease states). (G) Expression profiles of selected DEGs across disease states. (H) UMAP visualizations of extracted CD8[+] T cell barcodes, used for downstream single cell trajectory and classification analyses.

To ensure robustness and statistical reliability, all experiments were conducted 50 times, and the reported results represent the average performance across these runs. We trained the model on each simulated dataset and assessed its performance using multiclass classification metrics, with a particular focus on the Area Under the Receiver Operating Characteristic curve (AUROC) for each class (disease state). The ROC curves for each state in D1, D2, and D3 are shown in Fig 3A–3C, respectively. For instance, as depicted in Fig 3C, the model demonstrates high discriminative power across all four states in D3, with AUROC values of 1.00, 0.99, 0.98, and 1.00 respectively.

A summary of classification performance across the three simulation settings is presented in Table 1, which includes Accuracy, AUROC, Area Under the Precision - Recall Curve (AUPRC), F1-score, Precision, Recall, and Specificity for each dataset. Notably, classification accuracy increases as the number of genes rises, indicating that the method benefits from a richer feature space and scales effectively with network size.

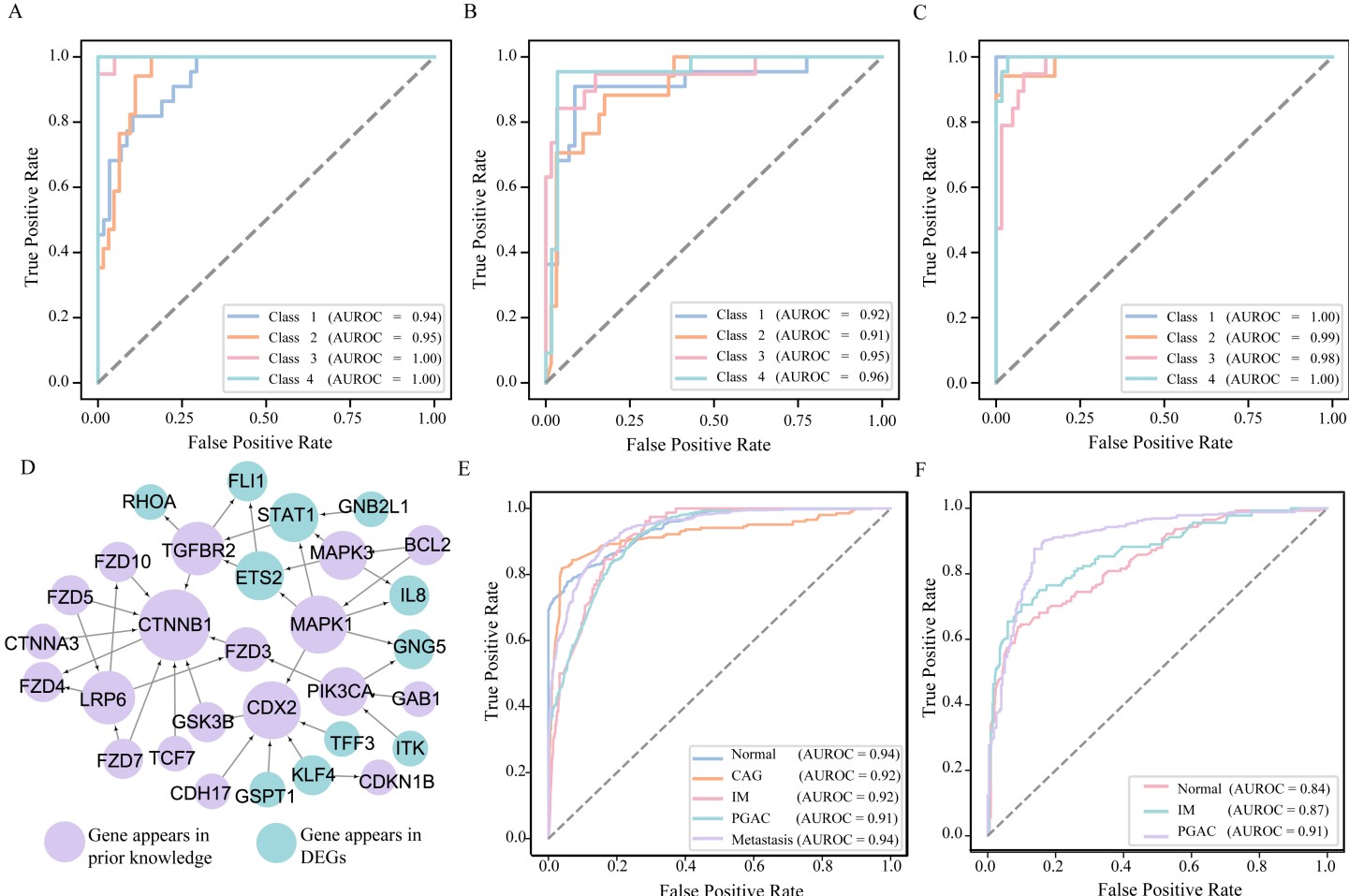

**Fig 3**. **classification results on simulated datasets and biomarkers and classification results for GAC dataset.** Each panel shows the ROC curves for four disease states in: (A) D1 with 30 genes, (B) D2 with 50 genes, and (C) D3 with 100 genes. (D) Dynamic Biomarkers for GAC. (E) ROC Curve for GAC dataset. (F) ROC curve for independent verification.

**Table 1**. Classification performance across the three simulation settings.

| Metric | 30 nodes | 50 nodes | 100 nodes |
|---|---|---|---|
| Accuracy | 0.8622 ± 0.0449 | 0.8921 ± 0.0402 | 0.9306 ± 0.0282 |
| AUROC | 0.9728 ± 0.0110 | 0.9358 ± 0.0214 | 0.9909 ± 0.0086 |
| AUPRC | 0.9275 ± 0.0290 | 0.8972 ± 0.0316 | 0.9759 ± 0.0178 |
| F1 Score | 0.8661 ± 0.0399 | 0.8528 ± 0.0502 | 0.9315 ± 0.0310 |
| Precision | 0.8420 ± 0.0380 | 0.8586 ± 0.0355 | 0.9299 ± 0.0298 |
| Recall | 0.8500 ± 0.0334 | 0.8481 ± 0.0369 | 0.9330 ± 0.0299 |
| Specificity | 0.9408 ± 0.0196 | 0.9167 ± 0.0127 | 0.9761 ± 0.0095 |

## Performance on gastric adenocarcinoma

To explore the cellular and molecular changes across different states of gastric adenocarcinoma (GAC), we utilized single cell RNA sequencing (scRNA-seq) datasets derived from multiple publicly available sources. These datasets capture the heterogeneity of the tumor microenvironment and offer a comprehensive view of the disease progression at single cell resolution.

In total, we used scRNA-seq data covering five distinct biological states: normal, chronic atrophic gastritis (CAG), intestinal metaplasia (IM), primary gastric adenocarcinoma (PGAC), and metastatic GAC. The precancerous states included CAG and IM samples, followed by malignant stages composed of primary GAC and metastatic lesions from sites such as the peritoneal cavity, ovary, and liver.

In this study, datasets GSE234129 [32] and GSE134520 [33] were used to train and test the proposed method for identifying dynamic biomarkers in GAC. An additional dataset, obtained from an institutional data portal (https://dna-discovery.stanford.edu) [34], was used for independent validation of the discovered dynamic biomarkers. After applying filtering and preprocessing steps, a total of 58,279 high quality single cells were retained for downstream analyses. The detailed description of these datasets and the computational tools used is provided in S1 Table.

To assess the effectiveness of the proposed method in real disease setting, we applied our framework to transcriptomic data of gastric adenocarcinoma covering five disease states: normal, CAG, IM, PGAC, and metastatic GAC. The model was trained on state-specific gene networks built using differentially expressed genes (DEGs) and prior regulatory knowledge. Graph embeddings from these networks were employed to identify disease-relevant dynamic network biomarkers (DNBs) and classify the disease states.

As shown in Fig 3D, our method identified 30 genes as state-associated DNBs for gastric cancer. These DNBs formed a connected regulatory subnetwork with 40 edges, based on curated interactions from RegNetwork (need the reference). Among the 30 identified genes, 19 overlapped with previously known GAC-related genes, while 11 were newly prioritized based on their differential expression profiles and topological prominence across states. This supports the model's ability to integrate prior biological knowledge with data-driven signals to discover both known and novel candidate biomarkers. The resulting DNB network captures key regulatory transitions associated with cancer progression.

To evaluate the discriminative power of the identified DNBs, we used the learned embeddings for multiclass classification of the five disease states. The model was evaluated over 50 independent runs and achieved strong performance across evaluation metrics, with an overall accuracy of 0.8755 ± 0.0341, AUROC of 0.9230 ± 0.0385, AUPRC of 0.8871 ± 0.0413, F1-score of 0.8607 ± 0.0357, precision of 0.8688 ± 0.0556, recall of 0.8686 ± 0.0286, and specificity of 0.8808 ± 0.0531. These results demonstrate that the selected DNBs provide robust representations capable of distinguishing subtle transitions between states of GAC progression.

Further analysis of classification performance across individual states, as illustrated by the AUROC values in Fig 3E, showed high predictive accuracy across all states: Normal (0.94), CAG (0.92), IM (0.92), PGAC (0.91), and Metastasis (0.94). These consistent AUROC scores highlight the model's ability to capture both early and late-state molecular signatures relevant to disease progression. To characterize the dynamic behavior of biomarker gene expression across disease

progression, we computed and visualized the state-specific mean expression profiles of the DNBs that did not overlap with previously known, as shown in S3 Fig.

To explore the functional relevance of the 30 selected DNB genes, we conducted pathway and gene ontology (GO) enrichment analysis using Metascape (http://metascape.org/) (results provided in the S4 Fig). The most significantly enriched pathway was gastric cancer (hsa05226), reinforcing the disease specificity of the identified gene set. In addition, several other pathways known to be involved in cancer development and progression were significantly enriched, including: GO:0030335 - Positive regulation of cell migration, GO:0048568 - Embryonic organ development, GO:0048762 - Mesenchymal cell differentiation, GO:0070848 - Response to growth factor, WP5036 - Angiotensin II receptor type 1 pathway, M223 - PID Beta Catenin Nuclear Pathway.

These pathways are closely linked to oncogenic processes such as epithelial-mesenchymal transition, angiogenesis, and cellular differentiation, all of which are known to contribute to GAC progression. The enrichment of these biological themes among the selected DNBs suggests that the proposed model not only achieves accurate classification but also uncovers functionally meaningful regulators of gastric tumor development. Detailed information on the identified DNBs, including their functional summaries, is provided in S6 Table.

### Independent verification

To further evaluate the robustness and generalizability of the identified DNBs, we assessed their classification performance on an independent dataset including three gastric adenocarcinoma states: Normal, IM, and PGAC. This external dataset, detailed in the Data section, was not utilized during training or model development and acts as an independent validation cohort.

TWhen the model was applied to this dataset, it exhibited strong generalization performance, achieving an accuracy of $0.8206 \pm 0.0576$, AUROC of $0.8738 \pm 0.0429$, AUPRC of $0.8091 \pm 0.0473$, F1-score of $0.7445 \pm 0.0636$, precision of $0.7365 \pm 0.0621$, recall of $0.8003 \pm 0.0741$, and specificity of $0.8168 \pm 0.0944$. These results emphasize the discriminative ability of the discovered DNBs in an unseen dataset and highlight their potential applicability across different clinical cohorts.

To further investigate state-specific classification effectiveness, we calculated AUROC values separately for each class. As shown in Fig 3F, the model achieved AUROC scores of 0.84 for Normal, 0.87 for IM, and 0.91 for PGAC. These findings confirm that the proposed framework can accurately identify key transitions even in datasets with limited state granularity, reinforcing the biological and predictive relevance of the selected DNBs across real-world gastric cancer datasets.

### Ablation experiments

To quantify the contributions of key components in the proposed TransMarker framework, we conducted a series of ablation experiments examining the effects of (i) topological feature integration, and (ii) rewiring strategies. The results are summarized in Fig 4, which collectively shows the impact of these elements on model performance and provides a comprehensive evaluation of metrics under different training conditions.

The first set of ablation experiments evaluates the contribution of local and global topological features to the final performance. As shown in Fig 4A, we tested four configurations: (1) using neither local nor global topological features (None), (2) using only global features (w/o local), (3) using only local features (w/o global), and (4) using both local and global features (our full model). The full model achieved the best performance across ACC, AUROC, and AUPRC evaluation metrics, confirming the complementary value of incorporating both local and global topological signals. Notably, excluding local features led to a performance drop, with accuracy of 0.796, AUROC of 0.841, and AUPRC of 0.784 ranking second overall. Omitting both feature types resulted in the most pronounced decline, underscoring the critical role of structural context in node representation learning. These findings validate the design of the topological integration module and

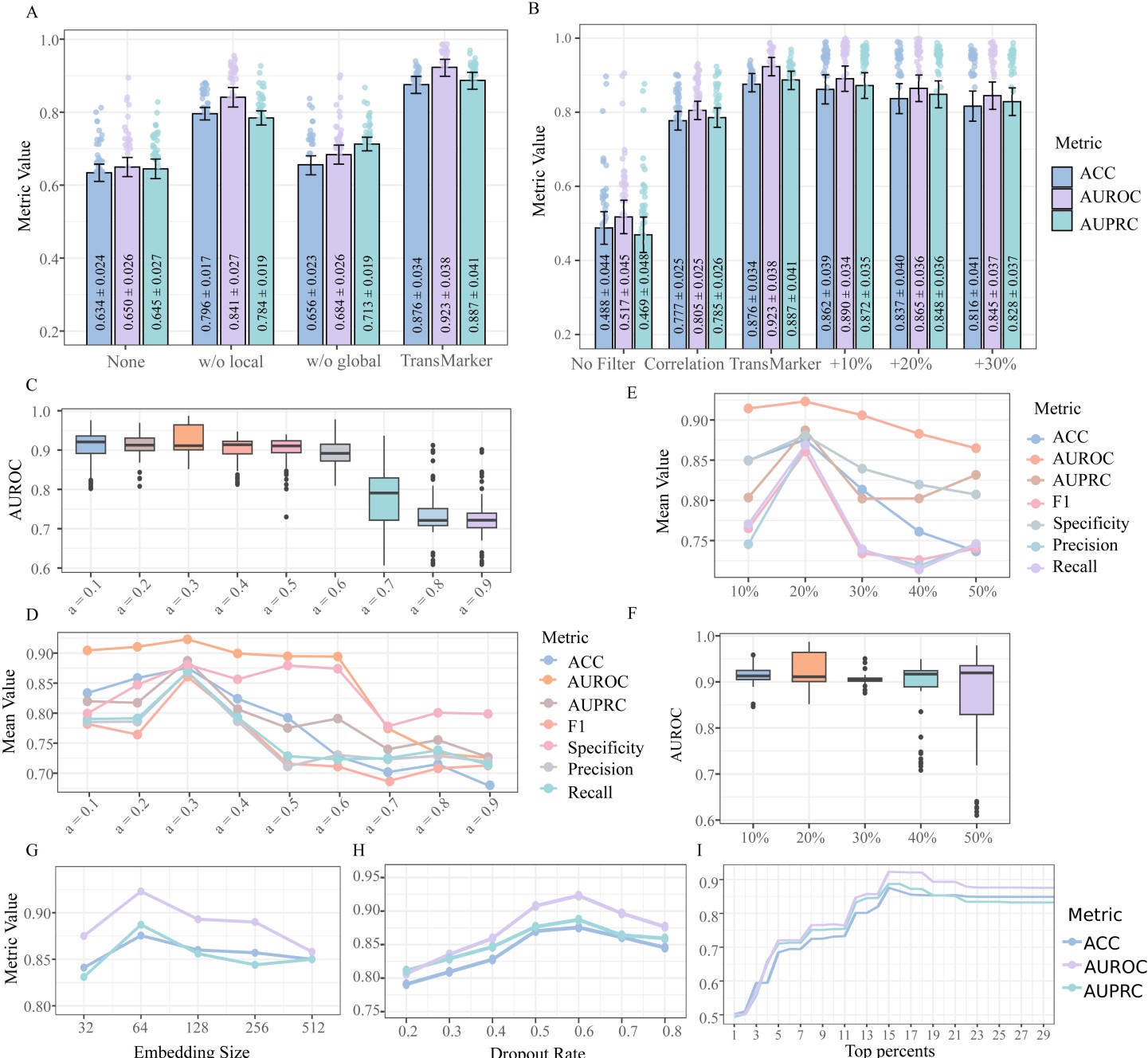

**Fig 4. Ablation studies and performance evaluation under varying training conditions.** (A) Performance comparison under four topological feature integration settings: no features (None), only global features (w/o local), only local features (w/o global), and both local and global features (Full). The full model achieves the best results across ACC, AUROC, and AUPRC metrics, confirming the complementary value of integrating both feature types. (B) Comparison of different network rewiring strategies: no rewiring, correlation-based rewiring, our proposed regulatory rewiring, and rewiring with random noise injection (10%, 20%, and 30%). Our method outperforms others, demonstrating both the importance of state-specific regulatory rewiring and the robustness of the approach under noisy conditions. (C–D) Sensitivity analysis of the balancing coefficient $\alpha$ that controls the contribution of local versus global topological features. Optimal performance is achieved at $\alpha$=0.3, highlighting the importance of a balanced integration. (E–F) Evaluation of model performance with varying training data sizes (10% to 50%). AUROC box plots (E) and full performance metrics (F) show that larger training sets yield more stable and accurate predictions. (G) Evaluation of model performance with different embedding dimensionalities (32 to 512) based on ACC, AUROC, and AUPRC. (H) Performance of the model under different dropout rates. (I) Classification performance using varying proportions (1% to 30%) of top-ranked genes; optimal performance is achieved using the top 15%.

emphasize that both local connectivity patterns and global graph structures are essential for accurate modeling of disease progression.

More detailed experiments investigate the sensitivity of the model to the balancing coefficient $\alpha$, which governs the relative contribution of local versus global topological features. As illustrated in Fig 4C and 4D, optimal performance was obtained at $\alpha = 0.3$, indicating that a slight emphasis on global features over local ones yields the best results.

Performance declined as $\alpha$ deviated from this optimum. Notably, when $\alpha$ approached 0.9, heavily favoring local features, model performance deteriorated significantly. This trend is consistent with the earlier observation (from Fig 4A) that global features contribute more substantially than local ones to predictive accuracy. These results confirm that the integration balance between local and global features is a key hyperparameter that must be carefully tuned to optimize performance. A detailed table of results across all evaluation metrics for the full range of $\alpha$ values is provided in the S2 Table.

The second group of experiments, visualized in Fig 4B, explores the effect of different network rewiring strategies on model performance. We compared four distinct conditions. In the first, no rewiring was applied, meaning all states shared the same regulatory network structure. The second condition involved correlation-based network construction, where state-specific co-expression networks were generated using Pearson correlation coefficients with a resampling-based thresholding strategy. The third condition implemented our proposed method, which constructs state-specific regulatory networks by integrating prior regulatory knowledge with dynamic expression signals. Lastly, to assess robustness to noise, we conducted additional experiments in which 10%, 20%, and 30% randomly added edges were injected into the regulatory network, simulating scenarios with incomplete or noisy prior knowledge.

The results demonstrate that our proposed rewiring method achieved the best classification performance across all evaluation metrics. In contrast, the no-rewiring condition produced the weakest performance, emphasizing the importance of modeling state-specific network dynamics. While correlation-based rewiring offered some improvement over the non-rewired setup, its performance was still lower, with AUROC dropping to approximately 0.80 and both accuracy and AUPRC falling below 0.80. Notably, the model maintained strong predictive capability under all three noise-injection scenarios, with accuracy above 0.81, AUROC above 0.84, and AUPRC above 0.82. These findings highlight the robustness of our approach and its potential applicability even in settings with noisy or incomplete regulatory information.

To determine the optimal training data proportion, we experimented with varying training set sizes ranging from 10% to 50%; the results are shown in Fig 4E (AUROC box plot), and Fig 4F (plots of all evaluation metrics) and S3 Table.

We evaluated the impact of different embedding dimensionalities (ranging from 32 to 512) on model performance using ACC, AUROC, and AUPRC metrics, as shown in Fig 4G. Fig 4H presents the performance across various dropout rates. Additionally, Fig 4I illustrates the classification performance when using different proportions (1% to 30%) of top-ranked genes, with 15% achieving the best results. Further robustness analyses are provided in the Supporting Information. Specifically, S2 Text and S7 Fig examine the robustness of TransMarker to the choice and completeness of the prior knowledge network, while S3 Text and S8 Fig analyze the influence of the entropic regularization parameter ($\varepsilon$) in the Gromov–Wasserstein alignment step.

## Case study on esophageal squamous cell carcinoma

To assess the applicability and generalizability of our proposed method beyond gastric adenocarcinoma, we performed one more case study on Esophageal Squamous Cell Carcinoma (ESCC) using transcriptomic data (GSE199654) [10] encompassing six pathological states: Normal, High-Grade Intraepithelial Neoplasia (HGIN), Inflammation (INF), Stage I, Stage II, and Stage III. This experiment aimed to determine whether the framework could successfully identify DNBs and distinguish disease progression states in a different cancer type.

Applying the same pipeline, we identified a set of 28 genes as state-associated DNBs for ESCC. Among these, 14 genes overlapped with prior knowledge, while the remaining 14 were newly prioritized differentially expressed genes

(DEGs), indicating the model's capacity to integrate known biology with data-driven discoveries. The corresponding DNB network is visualized in Fig 5A. Functional enrichment analysis of these genes further supports their involvement in cancer-related pathways, with detailed results provided in the S5 Fig.

The identified DNBs were employed for multiclass classification of the six ESCC states, yielding an overall ACC of 0.8451, AUROC of 0.9025, and AUPRC of 0.8602, indicating high classification performance. State-wise AUROC values, shown in Fig 5B, ranged from 0.85 to 0.94, demonstrating consistent predictive performance across both early and advanced disease states.

These results confirm the robustness and transferability of our method, highlighting its potential for broad application across different cancer types and progression models.

## Comparison with highly related methods

To rigorously evaluate the performance and generalizability of TransMarker in identifying disease-relevant DNBs, we compared it with five state-of-the-art node ranking methods specifically developed for multilayer networks, 13 widely used node ranking strategies based on classical centrality measures in complex networks, and two gene ranking frameworks: RL-GenRisk [21], which applies reinforcement learning to prioritize cancer genes in renal carcinoma, and DyNDG [20], which models gene expression dynamics through time-series multilayer networks for leukemia. These comparisons provide a comprehensive benchmark across both static and dynamic network analysis paradigms.

We constructed state-specific gene networks and extracted top-ranked genes using each method, selecting the same number of top genes as determined by our method's optimal selection threshold identified through preliminary experiments. The resulting gene subsets were then assessed through multiclass classification across disease states, and performance was evaluated across multiple standard metrics. To ensure a fair comparison, all methods, including ours, were run 50 times, and results were averaged across runs.

The node ranking methods for multilayer networks include multilayer entropy [35], ElementRank [36], Versatility Centrality [37], Versatility Degree Centrality [38], and Eigenvector Multicentrality [39]. Each of these methods captures distinct aspects of node importance across network layers and has been applied in various multilayer network analysis tasks. However, none of them is tailored for dynamic disease progression modeling or state-specific biomarker prioritization as addressed in our study. Each of these multilayer node-ranking methods captures distinct aspects of node importance

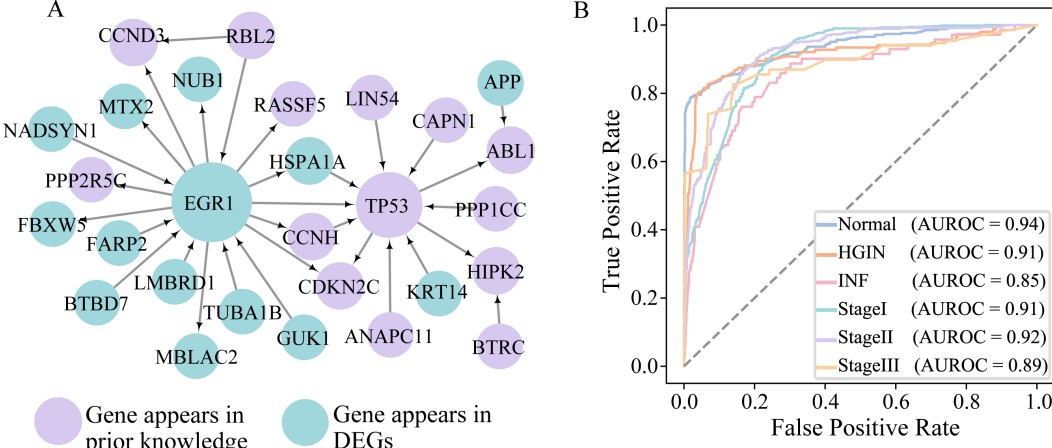

**Fig 5. Case study on Esophageal Squamous Cell Carcinoma (ESCC).** (A) State specific DNB subnetwork identified from GSE199654 across six ESCC states, including 28 genes (14 known, 14 novel). (B) State wise AUROC for multiclass classification using DNBs.

PLOS Computational Biology

across complex multilayer topologies. Specifically, Multilayer Entropy evaluates nodes based on structural information entropy, emphasizing those bridging different layers. ElementRank extends the PageRank algorithm by weighting layer-specific importance to reflect both local and global influence. Versatility Centrality employs a tensor-based approach to aggregate eigenvector centrality across layers, highlighting nodes consistently influential throughout the network. Versatility Degree Centrality integrates degree information from multiple layers to identify nodes with stable cross-layer connectivity. Eigenvector Multicentrality generalizes eigenvector centrality using a tensor model that simultaneously accounts for intra-layer and inter-layer propagation effects. These methods offer complementary perspectives on multilayer structure; however, unlike TransMarker, they do not explicitly capture dynamic regulatory rewiring or state-specific transitions using optimal transport principles, which are central to our framework. Details of the compared multilayer ranking methods, performance metrics, and the practical significance of evaluation criteria (ACC, AUROC, AUPRC, etc.) in the context of state-specific biomarker identification are provided in the S1 Text.

Table 2 summarizes the performance comparison on the GAC dataset. Our method consistently outperformed all others across all metrics. Among the competing methods, Versatility Degree Centrality achieved the highest AUROC after our method (0.8693 ± 0.0122), followed by ElementRank (0.8519 ± 0.0425). Notably, the performance gap in terms of AUPRC and F1-score was substantial, demonstrating the superior discriminative capacity of our DNBs. Methods such as Eigenvector Multicentrality and Versatility showed weaker performance, with accuracy below 0.5 and unstable precision-recall balance.

To further assess the distinctiveness and potential biological relevance of the genes prioritized by our approach, we compared the DNBs' genes selected by our method to those obtained from five multilayer network ranking methods in GAC using a Venn diagram analysis. As illustrated in Fig 6A, 19 of the 30 genes identified by TransMarker were not shared with any of the other methods, underscoring our method's ability to discover distinct molecular features that may play critical roles in the GAC progression process. Classification experiments using the expression profiles of these 30 genes across GAC states demonstrated strong discriminative performance, suggesting that the genes prioritized by our method are highly informative for distinguishing disease states.

Among the 19 uniquely identified genes, 7 (GSPT1, TFF3, KLF4, ITK, GNB2L1, FLI1, and GNG5) were not included in the KEGG gastric cancer pathway, indicating that they have not been previously recognized as canonical GAC-related genes. However, several of these genes have been implicated in tumor biology through independent studies. For instance, TFF3 has been associated with epithelial regeneration and cancer metastasis [40], and KLF4 is known for its dual role in tumor suppression and promotion depending on the cellular context [41]. GSPT1, a translation termination factor, has been reported to influence cell cycle regulation and proliferation in cancer [42]. The identification of these genes highlights our method's potential to uncover underexplored biomarkers that may contribute to GAC development or progression.

We also compared our method against 13 widely used node ranking strategies based on classical centrality measures in complex networks. These methods—including degree, betweenness, diffusion, PageRank, and eigenvector-based

**Table 2**. Quantitative performance comparison of our method and five multilayer network ranking baselines on the GAC dataset across multiple evaluation metrics.

| Method | Accuracy | AUROC | AUPRC | F1 Score | Precision | Recall | Specificity |
|---|---|---|---|---|---|---|---|
| TransMarker | 0.8755 ± 0.0341 | 0.9230 ± 0.0385 | 0.8871 ± 0.0413 | 0.8607 ± 0.0357 | 0.8688 ± 0.0556 | 0.8686 ± 0.0286 | 0.8808 ± 0.0531 |
| Multilayer Entropy | 0.5653 ± 0.0568 | 0.8319 ± 0.0096 | 0.5121 ± 0.0642 | 0.4983 ± 0.0390 | 0.5032 ± 0.0533 | 0.5656 ± 0.0451 | 0.8929 ± 0.0145 |
| ElementRank | 0.5626 ± 0.0321 | 0.8519 ± 0.0425 | 0.5415 ± 0.0422 | 0.5024 ± 0.0124 | 0.5035 ± 0.0322 | 0.5737 ± 0.0213 | 0.8915 ± 0.0183 |
| Versatility Centrality | 0.4587 ± 0.0144 | 0.7611 ± 0.0341 | 0.4028 ± 0.0123 | 0.3983 ± 0.0166 | 0.3993 ± 0.0143 | 0.4438 ± 0.0760 | 0.8602 ± 0.0681 |
| Versatility Degree Centrality | 0.6453 ± 0.0127 | 0.8693 ± 0.0122 | 0.6069 ± 0.0342 | 0.5566 ± 0.0130 | 0.5443 ± 0.0236 | 0.5819 ± 0.0572 | 0.8947 ± 0.0573 |
| Eigenvector Multicentrality | 0.4338 ± 0.0217 | 0.8019 ± 0.0150 | 0.5035 ± 0.0231 | 0.4432 ± 0.0224 | 0.4878 ± 0.0540 | 0.5157 ± 0.0243 | 0.8579 ± 0.0674 |

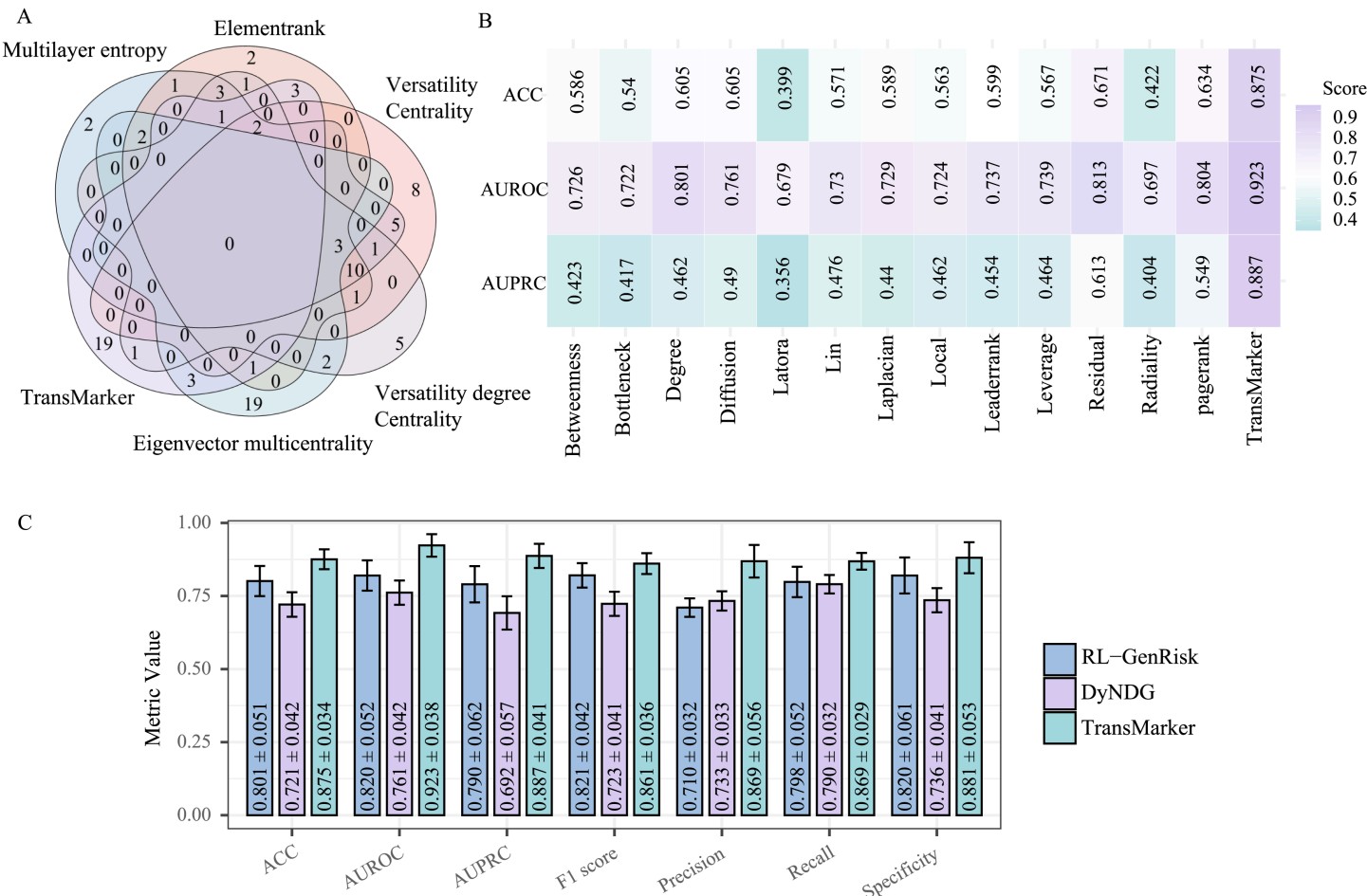

**Fig 6. Comparison of Gene Discovery and Performance Evaluation with Baseline Methods.** (A) Venn diagram comparing our method and five multilayer network baselines, highlighting 19 unique genes identified only by our approach, suggesting its potential to uncover novel GAC-related candidates. (B) Performance comparison between our method and 13 classical centrality-based ranking algorithms on the GAC dataset. (C) Performance comparison between our method and two gene ranking frameworks.

approaches—are commonly applied to biological networks to prioritize key nodes based on their topological properties. Each method captures a different aspect of network structure, such as node connectivity, shortest paths, local clustering, or influence propagation. The full list of methods and their mathematical definitions are provided in S5 Table. Fig 6B presents the comparative results of these 13 methods and our approach, evaluated using ACC, AUROC, and AUPRC on the GAC dataset. As shown, our method significantly outperforms all baselines across all three metrics, while most existing methods perform notably worse, with the best-performing alternative (PageRank) achieving only AUROC = 0.804 and AUPRC = 0.549. These results demonstrate the advantage of integrating both topological and biological signals in our framework. The comparison protocol follows the same evaluation strategy used in earlier benchmarking. Additional details of the performance metrics for all 14 methods are provided in S4 Table, and the overlap of selected genes is visualized in S6 Fig using an UpSet diagram.

To complement the benchmarking against classical and multilayer node ranking methods, we further evaluated our framework against two recent learning-based gene prioritization strategies, including RL-GenRisk and DyNDG—which have been designed for cancer-related gene identification in time series or multi-state contexts. As shown in Fig 6C,

our method (TransMarker) substantially outperforms both RL-GenRisk and DyNDG across all seven evaluation metrics, including accuracy, AUROC, AUPRC, F1-score, Precision, Recall, and Specificity. For instance, TransMarker achieves the highest AUROC (0.923 ± 0.0385) and AUPRC (0.8871 ± 0.0413), reflecting strong discriminative power and robustness in identifying state-relevant genes. In contrast, RL-GenRisk and DyNDG exhibit considerably lower AUPRC scores (0.790 and 0.692, respectively), suggesting limited precision-recall balance in multiclass classification. Similarly, TransMarker shows higher recall (0.8686 ± 0.0286) and F1-score (0.8607 ± 0.0357), indicating superior sensitivity and overall predictive performance. These results further demonstrate that our method not only captures the dynamic nature of disease progression more effectively than static or semi-dynamic models, but also generalizes well across diverse evaluation criteria. Taken together, the consistent advantage across all comparisons supports the robustness, adaptability, and clinical relevance of our framework for state-specific biomarker discovery in complex diseases such as GAC.

## Discussion

TransMarker is a graph-based framework designed to identify DNBs that signify state-specific transition within gene regulatory networks during disease progression. By integrating structural and expression information, it captures dynamic molecular signatures that mark critical states in disease development. The framework utilizes GATs to learn rich structural and contextual embeddings and aligns genes across layers to detect structural shifts in the embedding space. These shifts are then used to compute a DNI, enabling gene prioritization based on their dynamic behavior across disease states.

By incorporating these prioritized biomarkers, along with initial sample gene feature matrices, into a classification model, the framework not only achieves high accuracy in state prediction but also provides meaningful insights into disease dynamics and potential therapeutic targets.

To assess the utility of TransMarker, we conducted extensive experiments on three simulated datasets and three real-world single-cell datasets, including GAC (Gastric Adenocarcinoma) and ESCC (Esophageal Squamous Cell Carcinoma). Across all datasets, our model demonstrated robust performance in identifying biomarkers with significant discriminative power for multi-state classification. Results from ablation studies further validate the individual and collective contributions of each module to the overall effectiveness of our method.

Recent single-cell studies frequently model biological processes along continuous pseudo-time trajectories rather than discrete disease states. TransMarker could be adapted to such data by discretizing the pseudo-time continuum into ordered segments (e.g., early, middle, and late stages) and constructing corresponding regulatory networks for each segment. This would enable the framework to capture dynamic rewiring patterns and identify key transition regulators driving differentiation or disease progression along pseudo-temporal trajectories. Such an adaptation could provide valuable insights into the timing and sequence of molecular events underlying critical transitions. However, challenges include potential sensitivity to the pseudo-time inference method, variability in trajectory reconstruction across algorithms, and the need for principled segmentation strategies to balance temporal resolution and robustness. Future work will evaluate TransMarker on benchmark pseudo-time datasets to systematically assess its performance in reconstructing and prioritizing dynamic biomarkers in continuous single-cell processes.

From a computational perspective, TransMarker combines topological learning, GAT-based embedding, and Gromov–Wasserstein optimal transport alignment across disease states. The GAT component scales approximately linearly with the number of edges, while the GWOT step scales roughly with the square of the number of nodes but can be efficiently approximated through entropic regularization and mini-batch computation. When applied to the GAC dataset (five disease states, ~10,000 genes each), the complete training and alignment process took about 3 hours on a single NVIDIA Quadro RTX 6000 GPU (24 GB VRAM), with a peak memory footprint of ~13 GB. Since TransMarker operates on state-level networks and supports parallel processing across states, it remains computationally feasible and scalable for large multi-state disease analyses.

Although currently constrained by the limited availability of multi-state single-cell RNA-seq datasets, the modular and adaptable design of TransMarker makes it applicable to other diseases and data modalities as they become available.

In summary, TransMarker offers a principled and interpretable framework that captures dynamic changes in gene networks across disease states, paving the way for more accurate biomarker discovery and state-aware therapeutic strategies.

## Methods

### Single-cell RNA-seq data processing and cell type identification

To identify major immune cell types from the raw single-cell RNA-seq data, we implemented a systematic processing pipeline using the Seurat R package (v4.0) [43]. The raw count matrix was initially loaded into R and used to create a Seurat object with a minimum feature threshold of 200 genes per cell. Cells with fewer than 200 detected genes, more than 6,500 detected genes (to exclude potential doublets), or mitochondrial gene expression exceeding 15

Next, genes expressed in fewer than three cells were excluded to reduce noise and retain biologically relevant features. The filtered dataset was normalized, and 2,000 highly variable genes were identified. These variable genes were then used for downstream scaling and dimensionality reduction via principal component analysis (PCA).

For clustering, we constructed a shared nearest neighbor (SNN) graph and applied Louvain clustering with a resolution of 0.5, followed by UMAP for two-dimensional visualization of cell clusters. Known marker genes were utilized to aid in the identification of immune cell types.

Differential expression analysis was performed to identify marker genes for each cluster. Finally, the barcodes of CD8+ T cells were extracted and saved for further downstream analyses, with their UMAP visualizations presented in Fig 2H and S1 Fig.

### Differentially expressed genes identification and state-specific gene regulatory network construction

To delineate dynamic regulatory changes during gastric adenocarcinoma progression, we performed state-wise differential expression analysis and constructed corresponding gene regulatory networks. The detailed process for state-specific network construction is depicted in Fig 1A and 1B.

Differential expression analysis was carried out using the `edgeR` package in R. For each disease state, the matrix was compared to the normal state, and differentially expressed genes (DEGs) were identified using thresholds of $|\log_2 \text{ fold change}| > 1.0$ and $p$-value <0.05. Genes not consistently present in both conditions or showing missing or constant expression were excluded before modeling. The union of all DEGs identified across states was compiled, and expression matrices for each state were filtered to retain only the DEGs.

To enhance biological relevance, 150 gastric cancer-related genes from the KEGG [44] pathway database were added to the DEG set. This combined gene set was mapped onto a prior regulatory network sourced from RegNetwork [45], which integrates gene regulatory relationships from over 25 sources. The resulting state-independent background network comprised 2,363 genes and 6,798 directed regulatory interactions.

We then applied the PC-CMI algorithm [46] to eliminate indirect or spurious associations from the background network using expression data for each condition. This yielded five condition-specific gene regulatory networks (normal state and the four disease states), where nodes represent genes and directed edges represent condition-specific regulatory relationships supported by both prior knowledge and conditional mutual information. Details of the identified DEGs, edges across different states, and expression profiles of selected genes are shown in Fig 2E, 2F and 2G.

### The proposed method

Our model constructs a set of state-specific gene regulatory interaction graphs $\mathcal{G}^{(t)} = (\mathcal{V}, E^t)$ where $t \in \{1, 2, \dots, T\}$ denotes the disease state or time point. Each graph shares the same node set $\mathcal{V}$ (representing genes), with each node $v_i \in \mathcal{V}$

for $i = 1, 2, \ldots, N$, while the edge set $E^t$ captures state-dependent gene-gene relationships. By modeling these networks across states, the dynamic nature of disease progression is reflected in the evolving topologies. A graph attention network (GAT) is applied to each $\mathcal{G}^{(t)}$ to generate gene embeddings that integrate both topological features and node attribute information. These embeddings are then used to analyze expression and connectivity dynamics across states, facilitating the identification of dynamic network biomarkers (DNBs). Finally, these DNBs are employed to train a deep neural network for multi-class classification of disease states.

**Topological feature integration.** To enhance node representation with structural information, we integrate both local and global topological features derived from each state-specific graph $\mathcal{G}^{(t)}$. This integration produces a structural matrix that captures multi-scale relational properties among genes. Specifically, two types of topological descriptors are utilized: Local Topological Profile and Global Topological Profile.

To construct a unified representation that incorporates both local and global perspectives, we compute a convex combination of the two similarity matrices:

$$S_{ij}^{(t)} = \alpha \cdot S\_Local_{ij}^{(t)} + (1 - \alpha) \cdot S\_Global_{ij}^{(t)},$$

where $\alpha \in [0, 1]$ is a tunable hyperparameter controlling the balance between local and global influences. This combined structural matrix $S_{ij}^{(t)}$ is then employed as an adjacency matrix to enhance topological learning within the graph attention networks.

**Local topological profile.** The local topological structure of each graph $\mathcal{G}^{(t)}$ is captured using the shortest path distances between nodes. For each pair of nodes $v_i$ and $v_j$, the shortest path length $d_{ij}^{(t)}$ is calculated. To transform these distances into a similarity measure, we apply an exponential decay function:

$$S\_Local, ij^{(t)} = \exp\left(-d_{ij}^{(t)}\right).$$

If two nodes are disconnected in the graph, a large constant value is assigned to represent an infinite distance. The resulting matrix is then row-normalized to ensure comparability across nodes. This descriptor emphasizes local neighborhood proximity and captures how easily one gene can influence another through direct or short-path interactions.

**Global topological profile.** To capture the global structural importance of genes within each state-specific interaction network, we construct a Global Topological Profile. Unlike local structure, which emphasizes neighborhood proximity, the global profile reflects each gene's influence across the entire graph.

To quantify the global importance of each gene in the network, we compute PageRank scores $p_i^{(t)}$ for the graph $\mathcal{G}^{(t)}$. These scores estimate the long-range influence of each node throughout the graph. The global structural similarity between nodes $v_i$ and $v_j$ is then defined as the product of their PageRank scores:

$$S_{\text{Global},ij}^{(t)} = p_i^{(t)} \cdot p_j^{(t)}.$$

This formulation encodes pairwise global similarities such that genes with higher joint importance receive stronger weights. The resulting matrix is also row-normalized and captures global interaction potential by emphasizing nodes with central roles in the network's structure ($S_{\text{Global},ij}^{(t)} \in \mathbb{R}^{N \times N}$).

**State specific graph embedding module.** Our proposed model projects gene expression profiles along with the knowledge-based interaction matrix to a low-dimensional space. The objective is to optimize the embedding space to learn state-specific molecular representations from gene interaction networks, facilitating the identification of biomarkers linked to disease progression.

 

Given the gene expression matrix $X \in \mathbb{R}^{N \times M}$, where $N$ represents the number of genes and $M$ denotes the number of samples, along with an associated adjacency matrix $A^{(t)} \in \mathbb{R}^{N \times N}$ representing the gene topological network at disease state $t$, our model aims to learn a mapping function $f$ that generates low-dimensional representations $E_t \in \mathbb{R}^{N \times d}$ for state $t$:

$$E_t = f_\theta(X, A^{(t)}),$$

where $\theta$ represents the model parameters, and $d$ is the dimension of the latent space. These embeddings are utilized to uncover meaningful molecular patterns and identify state-specific biomarkers associated with disease progression.

Our model comprises two layers of Graph Attention Networks (GATs), applied independently to each state-specific graph. In the first GAT layer, we employ multi-head attention, allowing the model to capture diverse contextual relationships by aggregating information from multiple perspectives.

Let $h_i \in \mathbb{R}^M$ denote the initial feature vector of gene $i$, and $W^{(k)} \in \mathbb{R}^{M \times F}$ be the weight matrix for the $k$-th attention head. The attention coefficient $\alpha_{ij}^{(k)}$ for the interaction between gene $i$ and gene $j$ is computed as:

$$e_{ij}^{(k)} = a(W^{(k)} h_i, W^{(k)} h_j),$$

$$\alpha_{ij}^{(k)} = \frac{\exp(\text{LeakyReLU}(e_{ij}^{(k)}))}{\sum_{l \in \mathcal{N}_i} \exp(\text{LeakyReLU}(e_{il}^{(k)}))},$$

where $\mathcal{N}_i$ is the neighborhood of gene $i$ as defined by the state-specific network $A^{(t)}$; LeakyReLU is a non-linear activation function. In the first layer, the output for node $i$ is:

$$h_i' = \Big\Vert_{k=1}^{K} \sigma\left( \sum_{j \in \mathcal{N}_i} \alpha_{ij}^{(k)} W^{(k)} h_j \right),$$

where $\Vert$ denotes concatenation, $\sigma$ is the ELU activation function, and $K$ is the number of attention heads.

In contrast, the second GAT layer uses a single attention head to project the intermediate features into a unified embedding space. The layer can be formulated as:

$$h_i^{(2)} = \sigma\left( \sum_{j \in \mathcal{N}_i} \alpha_{ij} W h_j' \right),$$

Since our model is unsupervised, it does not rely on ground-truth labels. Instead, we design a contrastive objective to enforce embedding similarity between connected nodes. Specifically, we minimize the average Euclidean distance between the embeddings of node pairs connected in the adjacency matrix $A^{(t)}$, encouraging the model to preserve the network's structural topology in the latent space.

Alternatively, this can be seen as minimizing a reconstruction loss based on the similarity of node embeddings. The likelihood of an edge between node $i$ and node $j$ is computed as:

$$\hat{A}_{ij}^{(t)} = \sigma(e_i^T e_j),$$

where $e_i, e_j \in \mathbb{R}^d$ are the final embeddings of genes $i$ and $j$, and $\sigma$ is the sigmoid function. The binary cross-entropy loss between the reconstructed adjacency matrix $\hat{A}^{(t)}$ and the original $A^{(t)}$ is:

$$L_t = -\sum_{i,j} \left[ A_{ij}^{(t)} \log(\hat{A}_{ij}^{(t)}) + (1 - A_{ij}^{(t)}) \log(1 - \hat{A}_{ij}^{(t)}) \right].$$

The model is trained using the Adam optimizer, which adapts the learning rate for each parameter based on the first and second moments of the gradients, promoting faster and more stable convergence. We apply early stopping based on validation loss to prevent overfitting.

By training the GAT independently on each state-specific network, the model captures expression-informed topological embeddings that reflect key biological variations across disease progression. These embeddings are then analyzed to uncover potential biomarkers, focusing on genes whose network roles and expression profiles change significantly over time.

**Gromov-Wasserstein optimal transport for disease progression alignment.** To capture dynamic changes in gene-level structural relationships across disease progression, we employed Gromov-Wasserstein (GW) optimal transport, a method designed to match data distributions from different domains by exploiting their internal structural similarities, to align gene interaction embeddings between consecutive disease states. Unlike classical optimal transport, which assumes direct correspondence between elements, GW optimal transport (GWOT) identifies alignments based solely on the relational structure within each domain. Given that embeddings at each state encode structural gene-level features within the interaction network, GWOT enables alignment across states without requiring pointwise correspondences.

Let $D, D' \in \mathbb{R}^{N \times N}$ denote the pairwise dissimilarity matrices for two disease states, constructed from embedding representations of biological samples. GWOT seeks a probabilistic transport plan $\Gamma \in \mathbb{R}^{N \times N}$ that minimizes the discrepancy between the intra-domain structures by solving the following optimization problem:

$$\min_{\Gamma} \sum_{i,j,k,l} \left( D_{ij} - D'_{kl} \right)^2 \Gamma_{ik} \Gamma_{jl},$$
$$\text{s.t. } \Gamma \mathbf{1}_m = p, \ \Gamma^{\mathsf{T}} \mathbf{1}_n = q, \ \sum_{i,j} \Gamma_{ij} = 1,$$

where $p$ and $q$ are probability distributions over the source and target domains, respectively.

The resulting transport matrix $\Gamma$ can be interpreted as a soft matching between samples across different disease states. An entry $\Gamma_{ij}$ indicates the degree to which a sample in one state corresponds to a sample in the other, based on the similarity of their internal neighborhood structures.

To enhance computational efficiency and obtain smoother solutions, an entropy regularization term can be introduced. The entropic GWOT formulation adds a negative entropy term $H(\Gamma) = -\sum_{i,j} \Gamma_{ij} \log \Gamma_{ij}$ to the objective:

$$\min_{\Gamma} \left[ \sum_{i,j,k,l} (D_{ij} - D'_{kl})^2 \Gamma_{ik} \Gamma_{jl} - \varepsilon H(\Gamma) \right].$$

Here, the regularization parameter $\varepsilon > 0$ balances structure preservation and entropy maximization. Smaller values of $\varepsilon$ encourage sparser alignments, while larger values yield smoother transport plans. This formulation helps avoid poor local optima and improves alignment robustness.

Overall, GWOT provides a principled approach for discovering relationships between biological samples across multiple disease stages based on their intrinsic topological organization, without requiring prior alignment information.

To quantify the extent of structural variation experienced by each gene across states, we computed a cumulative alignment score. Specifically, for each gene, we summarized the standard deviations of its transport plan row vectors across all states transitions:

$$R_i = \sum_{t=1}^{T-1} \text{std}\left(\Gamma_{i,:}^{(t,t+1)}\right).$$

This cumulative alignment score reflects how consistently or variably a gene's relational context shifts as the disease progresses, enabling the identification of genes undergoing significant structural changes across states.

**Identification of biomarker candidates.** To identify dynamic biomarker modules linked to disease progression, structurally variable biomarker candidates are initially selected based on changes in their local network structure across states. Union subnetworks are then built using these genes, and their connected components are extracted. The dynamic behavior of each component is quantified by calculating the DNI across states, and its variability is evaluated using the standard deviation of DNI values. Components are subsequently ranked according to DNI variability across states, and the one with the highest standard deviation is chosen as the most dynamically disrupted subnetwork, representing a disease-relevant biomarker module.

The computed alignment scores are stored in a vector $\mathbf{R} = \{R_1, R_2, \ldots, R_N\}$, where $R_i$ represents the cumulative alignment score of gene $v_i$. An unsupervised filtering strategy is applied to identify structurally variable genes, utilizing the elbow method.

Let $\mathbf{R}_{\text{sorted}}$ be the alignment scores sorted in descending order, and let $\pi$ be the permutation indices such that $\mathbf{R}_{\text{sorted}} = \mathbf{R}_\pi$. The elbow point, indicating a significant shift in the slope of the ranking curve, is detected using the KneeLocator algorithm[47]. This point determines a threshold that separates highly variable genes from more stable ones. If the elbow is located at index $\delta$, the set of structurally variable genes is defined as:

$$\mathcal{V}_{\text{selected}} = \{v_{\pi_i} \mid i < \delta\}.$$

Genes ranked above the elbow are regarded as structurally variable across states. In cases where the elbow point is ambiguous or not well-defined, we adopt an empirically derived proportion determined through systematic examination to select the top-ranked genes. The final set $V_{\text{selected}}$ captures genes with dynamic topological behavior, serving as candidates for downstream disease progression analysis.

**Extraction of candidate components.** To identify biomarker modules undergoing dynamic structural changes across disease states, the analysis quantifies the variability of gene-gene interactions over time. This is achieved by computing the DNI, which captures abrupt topological changes within subnetworks.

A union network is constructed by merging all state-specific subnetworks induced from the previously selected structurally variable genes ($V_{\text{selected}} \subset V$). The union network is defined as:

$$\mathcal{G}^{(\text{union})} = (\mathcal{V}_{\text{selected}}, E^{\text{union}}),$$

where $E^{\text{union}} = \bigcup_{t=1}^{T} E^{(t)}|_{\mathcal{V}_{\text{selected}}}$. Here, $E^{(t)}|_{\mathcal{V}_{\text{selected}}}$ denotes the set of edges in state $t$ restricted to nodes in $\mathcal{V}_{\text{selected}}$.

From $\mathcal{G}^{(\text{union})}$, connected components are extracted:

$$\mathcal{C} = \{C_1, C_2, \ldots, C_k\},$$

where each $C_j \subseteq \mathcal{V}_{\text{selected}}$. These components serve as candidate gene modules. For each component $C_j$, its corresponding state specific subgraph is defined at each state $t$ as:

$$\mathcal{G}^{(t)}(C_j) = (C_j, E^{(t)}(C_j)),$$

where $E^{(t)}(C_j) = \{(u, v) \in E^{(t)} \mid u, v \in C_j\}$. These subgraphs are then used in subsequent steps to quantify the dynamic behavior of each component across states.

**Quantifying dynamic behavior via DNI variability.** To evaluate the structural and functional dynamics of each gene module during disease progression, we introduce a novel Dynamic Network Instability (DNI) metric. This metric measures the temporal variability of a module by assessing changes in both alignment uncertainty and network cohesion across states. For a connected component $C_j$, the DNI is defined as:

$$\text{DNI}_{C_j} = \sigma\left(D_{C_j}^{(1)}, D_{C_j}^{(2)}, \ldots, D_{C_j}^{(t)}\right),$$

where $D_{C_j}^{(t)}$ is the state-specific dynamic score of component $C_j$, calculated as:

$$D_{C_j}^{(t)} = \exp\left(-\text{GCS}_{C_j}^{(t)}\right).$$

The Graph Cohesion Score (GCS) is a composite term defined as:

$$\text{GCS}_{C_j}^{(t)} = \left(1 - \rho_{C_j}^{(t)}\right) \cdot \left(-\Psi_{C_j}^{(t)}\right),$$

where $\Psi_{C_j}^{(t)} = \sum_{i \in C_j} R_i^{(t)}$ is the sum of alignment scores for nodes in $C_j$, reflecting the heterogeneity of node alignment within the component.

$\rho_{C_j}^{(t)}$ is the cohesion density, defined as the edge density of the component:

$$\rho_{C_j}^{(t)} = \frac{2m_t}{n_t(n_t - 1)},$$

with $n_t$ and $m_t$ representing the number of nodes and edges in the component $\mathcal{G}^{(t)}(C_j)$, respectively.

This formulation ensures that modules with high internal connectivity and strong overall alignment yield lower GCS values, leading to higher $D_{C_j}^{(t)}$ scores. All components $C_j \in \mathcal{C}$ are ranked by their DNI variability:

$$C^* = \arg\max_j \text{DNI}(C_j).$$

The selected $C^*$ represents the most temporally dynamic module, exhibiting the highest variation in structure across states.

**Classification method.** To further assess the predictive capacity of the identified biomarker module during disease progression, we devise a classification framework specific to multi-state gene expression data.

We employ a multi-layer perceptron (MLP) neural network to classify samples across different states using the expression profiles of biomarker genes. The expression data, initially organized by state, is concatenated after filtering to retain only biomarker-related genes. Each state-specific sample is assigned a one-hot encoded label indicating its corresponding state. To ensure a robust evaluation, we report multiple performance metrics, including classification accuracy, AUROC (macro-average over all states), AUPRC, precision, recall, F1-score, and specificity. These comprehensive metrics enable us to evaluate the discriminative power of biomarker gene expression in distinguishing among states, reflecting their relevance in capturing dynamic transcriptomic changes over time.

## Implementation details

In the embedding process, the GAT model comprised two attention layers with a dropout rate of 0.6, using 64 hidden units and 2 attention heads in the first layer, followed by a 62-dimensional output layer. The model was trained using the Adam optimizer [48] with a learning rate of 0.001 and weight decay of 5e-4. A contrastive loss function was employed to enforce similarity between embeddings of connected nodes. Early stopping was implemented to terminate training if the loss did not improve for 50 consecutive epochs. Parameter choices were further refined through experiments on dropout rates and embedding sizes.

Experiments were also conducted to identify the optimal proportion of top-ranked genes for downstream classification. By systematically evaluating multiple candidate percentages, we determined that selecting the top 15% of genes consistently delivered the best classification performance across datasets. This empirically determined selection ratio was subsequently adopted for all methods to ensure fair and consistent comparison.

To assess the state-discriminative power of the identified biomarker genes, a multiclass classification model was implemented using an MLP. Expression values across all disease states were concatenated to form the input feature matrix. One-hot encoded labels were assigned to each sample based on its state. The dataset was divided into 70% training, 10% validation, and 20% test sets, and class weights were computed to address any class imbalance.

The MLP architecture consisted of three fully connected layers with 128, 64, and 32 units, respectively, each using ReLU activation. The final output layer comprised 5 units with a Softmax activation function to represent the five states. The model was trained using categorical cross-entropy loss and evaluated using multiple performance metrics. Early stopping was applied based on validation AUROC, with a patience of 50 epochs, and the best-performing model weights were retained for final evaluation.

## Complexity analysis

The proposed framework combines topological learning, state specific embedding, optimal transport-based alignment, and biomarker module identification across disease states. Its overall time complexity is primarily determined by three core computational components: (1) Topological Feature Integration, (2) State-specific GAT-based Embedding, and (3) GWOT for dynamic alignment.

In the Topological Feature Integration step, for each state-specific network $\mathcal{G}^{(t)}$, computing the Local Topological Profile requires all-pairs shortest path distances, with a time complexity of $\mathcal{O}(N^2 \log N + N|E^{(t)}|)$ for a sparse graph with $N$ nodes and $|E^{(t)}|$ edges. The Global Topological Profile is derived via PageRank, which has an average-case complexity of $\mathcal{O}(|E^{(t)}|)$ per iteration and typically converges in a small number of steps. The convex combination and normalization of similarity matrices add negligible overhead $\mathcal{O}(N^2)$, resulting in an overall time complexity per state of $\mathcal{O}(N^2 \log N + N|E^{(t)}|)$.

The State-Specific Graph Embedding Module consists of two layers of GATs. In the first layer, multi-head attention is applied with $K$ heads, each aggregating features from neighboring nodes. Given $d$ as the embedding dimension and $M$ as the input feature dimension, the per-layer complexity is $\mathcal{O}(K|V|dMv|E^{(t)}|d)$. Since the second layer uses a single head and operates on the concatenated output, the total embedding complexity per state becomes $\mathcal{O}(K(|V|dM + |E^{(t)}|d))$. This step is repeated for each time point $t \in \{1, 2, \ldots, T\}$, and since each graph is processed independently, the embedding step is inherently parallelizable.

In the GWOT module, the alignment between each pair of consecutive states involves solving a quadratic optimization over the dissimilarity matrices of size $N \times N$, resulting in a theoretical complexity of $\mathcal{O}(N^4)$. However, by introducing entropic regularization and using Sinkhorn-like iterative solvers, the practical complexity is significantly reduced to approximately $\mathcal{O}(N^2 \log N)$ per pair of states. For $T-1$ consecutive state transitions, the cumulative cost becomes $\mathcal{O}((T-1)N^2 \log N)$.

The Biomarker Identification and Component Extraction steps are dominated by sorting operations and connected component extraction. Sorting the cumulative alignment scores takes $\mathcal{O}(N \log N)$, and connected component detection from union subnetworks is $\mathcal{O}(|V| + |E^{union}|)$, which is negligible compared to earlier steps.

In summary, the overall time complexity of the proposed model is:

$$\mathcal{O}\left(T(N^2 \log N + N|E^{(t)}| + K(|V|dM + |E^{(t)}|d)) + (T-1)N^2 \log N\right).$$

This expression reflects the model's dependence on the number of disease states $T$, the network size $N$, the number of edges $|E^{(t)}|$, and embedding dimensionality $d$. Importantly, both the embedding and alignment components are highly parallelizable, making the model efficient and scalable for large-scale, state-resolved biological networks.

## Supporting information

**S1 Table. Details of datasets and tools.**
(PDF)

**S1 Fig. UMAP visualizations of extracted CD8$^{+}$ T and CD4$^{+}$ T cell barcodes, employed for downstream subsequent single-cell trajectory and classification analyses.**
(EPS)

**S2 Fig. Gene regulatory networks for different states of GAC post-rewiring.**
(EPS)

**S3 Fig. Mean expression levels of biomarker genes across various disease states.** The line plot illustrates the statewise mean expression of 11 biomarker genes identified as differentially expressed (DEGs) in the study. To improve clarity and minimize visual clutter, only the DEGs among the chosen biomarkers are shown—genes exclusively from the KEGG gastric cancer pathway are excluded from this figure.
(EPS)

**S4 Fig. Functional enrichment analysis of the DNBs identified for GAC.** The enriched pathways and GO terms exhibit a strong association with gastric cancer (hsa05226) and other cancer-related processes, including cell migration, embryonic organ development, mesenchymal cell differentiation, and response to growth factor signaling.
(EPS)

**S5 Fig. Functional enrichment analysis of the 28 DNB genes identified for ESCC.** The analysis underscores significant enrichment in multiple cancer-associated pathways.
(EPS)

**S2 Table. Detailed evaluation metrics for various $\alpha$ values.** Performance outcomes for all metrics across the complete range of $\alpha$ values utilized in the Topological Feature Integration step.
(PDF)

**S3 Table. Summary of evaluation metrics across different training data proportions.** It enumerates average performance scores for each metric across varying training set sizes (10%–50%).
(PDF)

**S1 Text. Details of Highly Related Methods and Evaluation Metrics.**
(PDF)

**S4 Table. Summary of performance metrics for our method and 13 centrality-based node ranking methods on the GAC dataset.** Evaluation encompasses accuracy, AUROC, AUPRC, and other standard metrics computed over 50 independent runs.
(PDF)

**S6 Fig. UpSet diagram illustrating the overlap of DNBs selected by our method and top-ranked genes by the 13 baseline centrality-based ranking algorithms.** The diagram reveals limited consensus among methods, with our method pinpointing a distinct subset of genes not commonly selected by other approaches. This indicates the uniqueness of our ranking strategy and its potential to uncover novel biomarkers in GAC.
(EPS)

**S5 Table. List of 13 centrality methods utilized for comparative study.**
(PDF)

**S6 Table. The dynamic network biomarkers based on TransMarker, a long with the summaries of their functions.**
(PDF)

**S2 Text. Robustness of TransMarker to the selection and completeness of the prior knowledge network.**
(PDF)

**S7 Fig. Robustness of TransMarker concerning the selection and completeness of the prior knowledge network.** Performance comparison of TransMarker when utilizing different prior interaction networks (HumanNet, InBioMap, STRINGdb) and under progressive random edge removal (−5%, −10%, −15%, and −20%) from the original RegNetwork prior. Bars represent mean $\pm$ standard deviation across multiple runs for ACC, AUROC, and AUPRC.
(EPS)

**S3 Text. Robustness to the Entropic Regularization Parameter in the Gromov–Wasserstein Alignment.**
(PDF)

**S8 Fig. Robustness analysis of the entropic regularization parameter in the Gromov–Wasserstein alignment step of TransMarker.** (A) Classification performance (ACC, AUROC, and AUPRC) across $\varepsilon \in \{1e-3, 5e-3, 1e-2, 5e-2, 1e-1\}$. (B) Spearman rank correlation of gene prioritizations relative to the default $\varepsilon = 1e-2$. (C) Mean rank deviation of the top 20 biomarkers across $\varepsilon$ values.
(EPS)

## Author contributions

**Conceptualization:** Zhi-Ping Liu.

**Data curation:** Fatemeh Keikha, Chuanyuan Wang, Zhixia Yang.

**Formal analysis:** Fatemeh Keikha, Zhi-Ping Liu.

**Funding acquisition:** Zhi-Ping Liu.

**Methodology:** Fatemeh Keikha, Zhi-Ping Liu.

**Project administration:** Fatemeh Keikha, Zhi-Ping Liu.

**Supervision:** Zhi-Ping Liu.

**Validation:** Fatemeh Keikha, Chuanyuan Wang, Zhi-Ping Liu.

**Visualization:** Fatemeh Keikha, Chuanyuan Wang.

**Writing – original draft:** Fatemeh Keikha.

**Writing – review & editing:** Fatemeh Keikha, Chuanyuan Wang, Zhixia Yang, Zhi-Ping Liu.

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
