## [Decision Letter · Decision Letter 0]

2 Oct 2025

PCOMPBIOL-D-25-01660

TransMarker: Unveiling Dynamic Network Biomarkers in Cancer Progression Through Cross-State Graph Alignment and Optimal Transport

PLOS Computational Biology

Dear Dr. Liu,

Thank you for submitting your manuscript to PLOS Computational Biology. After careful consideration and reviewing by three experts, we feel that the manuscript has a meaningful topic, but needs further improvements according to reviewers' comments. Therefore, we invite you to submit a revised version of the manuscript that addresses the points raised during the review process.

Please submit your revised manuscript within 30 days Dec 02 2025 11:59PM. If you will need more time than this to complete your revisions, please reply to this message or contact the journal office at ploscompbiol@plos.org. Please include the following items when submitting your revised manuscript:

We look forward to receiving your revised manuscript.

Kind regards,

Jifan Shi

Academic Editor

PLOS Computational Biology

Stacey Finley

Section Editor

PLOS Computational Biology

**Journal Requirements:**

At this stage, the following Authors/Authors require contributions: Fatemeh Keikha, Chuanyuan Wang, Zhixia Yang, and Zhi-Ping Liu. Please ensure that the full contributions of each author are acknowledged in the "Add/Edit/Remove Authors" section of our submission form.

6) Please ensure that the funders and grant numbers match between the Financial Disclosure field and the Funding Information tab in your submission form. Note that the funders must be provided in the same order in both places as well. 

**Reviewers' comments:**

Reviewer's Responses to Questions

**Comments to the Authors:**

**Please note that one **
**review is uploaded as an attachment.**

Reviewer #1: In this manuscript, the authors present a novel computational framework TransMarker for identifying dynamic biomarkers of cancer progression using single-cell RNA-seq data. The central idea is that cancer evolution involves not only changes in expression levels but also rewiring of gene regulatory networks across disease states. The framework integrates prior regulatory knowledge with state-specific gene expression to construct GRNs, where Graph Attention Networks (GATs) are used to learn contextual embeddings for each state, and Gromov-Wasserstein optimal transport quantifies structural shifts in gene roles across states. The significant changed genes are then used for disease state classification as DNBs. The authors validate TransMarker through both synthetic simulations and real-world datasets such as GAC and ESCC with systematic benchmarking and rigorous ablation studies, as well biological interpretability analysis.

Overall, this is a strong submission and solid work, and the manuscript is very well-written. I suggest the acceptance of manuscript by PlosCB. I have only one minor suggestion for authors to consider if that’s appropriate: Recently, there are several dynamical OT based methods to analyze scRNA-seq data (e.g. TIGON, NMI 2024; TrajectoryNet,ICML 2020; DeepRUOT, ICLR 2025; srVCR, Biorxiv 2024), and inference of GRN is also possible from these approaches. The authors might discuss these related works in the intro/discussion of the manuscript to further enhance the interests for general audience.

Reviewer #2: The manuscript presents TransMarker, a multilayer network framework that integrates prior gene–gene interactions, state-specific single-cell expression, and Gromov–Wasserstein optimal transport to identify dynamic network biomarkers across cancer progression. The method is well motivated, the experiments are thorough, and the results convincingly demonstrate superiority over existing multilayer ranking approaches. The paper is generally well written and of interest to the computational biology community. I listed here some comments and presentation suggestions to further improve clarity and reproducibility:

1. Framework Figure (Figure 1): In panel 1-E (Graph Alignment step), the role of optimal transport is central but not explicitly illustrated. Please modify the figure to depict the Gromov–Wasserstein optimal transport step so that the reader can visually follow this critical component.

2. Optimization Problem Formulation: After line 591, where the Gromov–Wasserstein distance is introduced, please provide the explicit standard mathematical form of the optimization problem. This will help readers reproduce the method without consulting external sources

3. Figure Font Consistency: Ensure consistent and sufficiently large font sizes across all figures. In particular, Figures 2, 3, 4, and 6 contain axis labels and legends that are noticeably smaller than in other panels, making them difficult to read.

4. Clarification of Comparative Methods: In the “Comparison with highly related methods” section, please expand the discussion of the five multilayer node-ranking methods used for benchmarking. For each, briefly explain which aspects of node importance across multilayer topologies they capture. This contextualization will help readers understand why these particular methods were chosen and how their design principles differ from TransMarker’s integration of regulatory rewiring and optimal transport.

5. Typographical consistency: There are several instances of inconsistent naming of the compared methods. For example, the same method appears with different capitalization and occasional extra characters (e.g., a dash or underscore) in Table 2 versus Figure 6A. Please standardize the naming convention for all methods across the main text, tables, figure panels, and captions. Ensure that each method’s name is written exactly the same where it is introduced, discussed in the Results, and labeled in figures. In particular, correct the mismatch between Table 2 and Figure 6A. Adopting a uniform style will improve clarity.

6. Some most recent computational methods for identifying the critical states based on the dynamic network biomarker (DNB) or its modifications are missing. I suggest the following recent and relevant literatures should be addressed:

- Earthquake alerting based on spatial geodetic data by spatiotemporal information transformation learning. Proceedings of the National Academy of Sciences, USA, 2023, 120(37):e2302275120.

- Ultralow-dimensionality reduction for identifying critical transitions by spatial-temporal PCA. Advanced Science, 2025, 9:2408173.

- sPGGM: a sample-perturbed Gaussian graphical model for identifying pre-disease stages and signaling molecules of disease progression. National Science Review, 2025: nwaf189.

- Uncovering the pre-deterioration state during disease progression based on sample-specific causality network entropy (SCNE). Research, 2024, 7:0368.

Reviewer #3: Please see the attached file.

**Have the authors made all data and (if applicable) computational code underlying the findings in their manuscript fully available?**

Reviewer #1: Yes

Reviewer #2: None

Reviewer #3: None

PLOS authors have the option to publish the peer review history of their article (what does this mean?). If published, this will include your full peer review and any attached files.

Reviewer #1: No

Reviewer #2: No

Reviewer #3: No

**Figure resubmission:**
---

## [Decision Letter · Decision Letter 1]

13 Nov 2025

Dear Dr. Liu,

We are pleased to inform you that your manuscript 'TransMarker: Unveiling dynamic network biomarkers in cancer progression through cross-state graph alignment and optimal transport' has been provisionally accepted for publication in PLOS Computational Biology.

Best regards,

Jifan Shi

Academic Editor

PLOS Computational Biology

Stacey Finley

Section Editor

PLOS Computational Biology

Reviewer's Responses to Questions

**Comments to the Authors:**

Reviewer #2: All my concerns have been addressed.

Reviewer #3: Thank you for the thorough revisions. All previous comments have been addressed. I have no further concerns.

**Have the authors made all data and (if applicable) computational code underlying the findings in their manuscript fully available?**

Reviewer #2: None

Reviewer #3: None

PLOS authors have the option to publish the peer review history of their article (what does this mean?). If published, this will include your full peer review and any attached files.

Reviewer #2: No

Reviewer #3: No

---

## [Editor Report · Acceptance letter]

PCOMPBIOL-D-25-01660R1

TransMarker: Unveiling dynamic network biomarkers in cancer progression through cross-state graph alignment and optimal transport

Dear Dr Liu,

I am pleased to inform you that your manuscript has been formally accepted for publication in PLOS Computational Biology. Your manuscript is now with our production department and you will be notified of the publication date in due course.

With kind regards,

Anita Estes
